# Interactions between Ewes and Rams during Mating Can Be Used to Predict Lambing Dates Accurately, but Not Sire

**DOI:** 10.3390/ani12131707

**Published:** 2022-07-01

**Authors:** Kirsty Cunningham, Andrew Van Burgel, Khama R. Kelman, Claire M. Macleay, Beth L. Paganoni, Andrew N. Thompson

**Affiliations:** 1Department of Primary Industries and Regional Development, 453 Katanning-Nyabing Rd, Katanning, WA 6317, Australia; kirsty.cunningham@dpird.wa.gov.au; 2Department of Primary Industries and Regional Development, 444 Albany Highway, Albany, WA 6330, Australia; andrew.vanburgel@dpird.wa.gov.au; 3Centre for Animal Production and Health, Murdoch University, Murdoch, WA 6150, Australia; k.kelman@murdoch.edu.au (K.R.K.); andrew.thompson@murdoch.edu.au (A.N.T.); 4Department of Primary Industries and Regional Development, 1 Verscheur Place, Bunbury, WA 6230, Australia; claire.macleay@dpird.wa.gov.au

**Keywords:** lamb survival, sensors, precision management, predicting oestrus, lambing date

## Abstract

**Simple Summary:**

The early prediction of lambing dates for ewes could improve their management at critical time points during pregnancy and throughout lambing. A pilot study used proximity sensors to measure the interactions between ewes and rams during mating to predict conception and lambing dates successfully. This study is an extension of the pilot study using multiple rams under commercial conditions. Lambing dates were predicted based on peak ewe–ram interactions during mating and then compared to the actual lambing dates observed in the paddock. Overall, 91% of lambing dates were predicted within ±6 days and 84% within ±3 days. The use of proximity sensors during mating was an effective means of predicting lambing dates. This could help improve ewe management during pregnancy and lambing and the subsequent survival and performance of their lambs.

**Abstract:**

Ewes often lamb over extended periods so the level of nutrition during pregnancy and lambing may be suboptimal for ewes that conceived later during mating. Predicting lambing dates would allow cohorts of ewes with similar gestational ages to be managed more precisely to achieve targets for ewe nutrition, feed on offer, mob sizes and access to shelter to improve lamb survival. The interactions between ewes and rams during mating have been used to predict the time of oestrus and lambing dates successfully, but this has not been tested at a commercial scale. In this study, proximity sensors were used to measure interactions between inexperienced Merino ewes (*n* = 317) and experienced rams (*n* = 9) during a 27-day mating period under commercial production conditions. When the gestation length was assumed to be 150 days, 91% of lambing dates were predicted within ±6 days of the actual birth date of lambs and 84% of lambing dates were predicted within ±3 days. The use of proximity sensors during mating was an effective means of predicting lambing dates, and there was no significant difference in accuracy for single bearing verses multiple bearing ewes. However, DNA parentage data showed the ram corresponding with the maximum daily interactions ratio was the sire for only 16% of all progeny, suggesting they could not be used to indicate the sire of the progeny.

## 1. Introduction

Lamb mortality is a significant economic and welfare issue for sheep producers in Australia and other major sheep producing countries. In Australia, about 25% of all lambs born will die, with over 80% of deaths occurring within three days after birth and mortality rates being higher for multiple rather than single born lambs [1,2,3]. Lamb mortality can be reduced by improving ewe nutrition during pregnancy to optimize lamb birth weights [4,5,6] and/or increasing the feed on offer [1], reducing mob size [7,8] and providing shelter [9,10] during lambing. The impact of these strategies on reducing mortality rates are greater for multiple born than single born lambs. Hence, adoption of pregnancy scanning for foetal number to enable differential management of single and multiple bearing ewes is critical to improving survival [11,12]. Reducing lamb mortality using these strategies on extensively grazed farms is often constrained by limitations to feed supply, large paddock sizes, access to sheltered paddocks suitable for lambing and the spread of lambing dates.

A recent survey of 1200 Merino producers across Australia indicated that the average mating period was almost 8-weeks, and only 20% of producers adopted the recommended mating period of 5 weeks or less [13,14,15]. It is often difficult to maintain ewe condition score and feed on offer targets over these extended lambing periods, so the condition and lambing environment may be suboptimal for ewes that conceived later during mating. Accurate prediction of lambing dates would allow cohorts of ewes with similar gestational ages to be managed judiciously to achieve targets for ewe nutrition, feed on offer, mob sizes and access to shelter to improve lamb survival.

Transabdominal ultrasound scanning can be used to predict differences in gestational age, but published studies on the accuracy of commercial ultrasound scanning to predict gestational age are scarce. Estimates of gestational age to within one oestrus cycle vary in accuracy from 56 to 100% (14–15; Paganoni unpublished data). The use of ultrasound techniques to estimate fetal age requires highly skilled technicians and the accuracy achieved is dependent on the technician [16]. Actual lambing date is also influenced by gestation length, which can vary by up to 10-days between individual ewes [17,18]. Bunter et al. [15] demonstrated that actual lambing date could be predicted within ±7 days with an accuracy of around 90% using a highly competent ultrasound technician. Similarly, Robertson et al. [19] reported that between 86 and 94% of ewes lambed within ±5-days of the predicted lambing date when the foetuses were 10.5 to 12 weeks old at scanning; however, when the foetuses were 8.5 to 10 weeks old the accuracy dropped to 23 to 44%. Furthermore, increasing litter size reduces the accuracy of these estimates due to reduced visibility and increased variation in fetus size between litter mates [20,21]. More accurate and readily adoptable methods of predicting lambing dates require investigation.

On-animal sensor systems that measure changes in ewe–ram behaviours and interactions during the mating period offer potential for improving the prediction of lambing dates. Alhamada et al. [22,23] used electronic devices to record when rams mounted ewes to detect time of oestrus. These devices used were deemed inappropriate for monitoring oestrus in large numbers of ewes under extensive grazing conditions [24]. O’Neill et al. [25] used proximity sensors to demonstrate significant increases in the number and duration of interactions between a single bull and cows during oestrus. Likewise, Paganoni et al. [24] confirmed that an increase in ewe–ram interactions was an effective indicator of conception date that could be used to accurately predict lambing dates. This work [24] involved small numbers of adult Merino ewes (29 or 42) exposed to a single ram in small paddocks (1 ha) for 26-days. This current study aims to predict lambing dates using proximity sensors for larger numbers of ewes joined with multiple rams on a larger paddock that is more typical of commercial production conditions. We reasoned that by measuring the interactions between ewes and rams during mating, as an indicator of time of oestrus, we can accurately predict the lambing dates of ewes under commercial production conditions.

## 2. Materials and Methods

### 2.1. Animals

Four hundred and sixteen maiden Merino ewes (born 2019) were mated with nine adult (2016-drop, experienced, testicles in-tact) Merino rams in 2021 on a dry cereal stubble (50 ha) for 35 days at the Katanning Research Facility (33.69° S and 117.57° E). Daily temperature and rainfall were recorded by a weather station at the facility. All ewes and rams were fitted with proximity sensors and grazed together over the mating period. After 35 days together ewes and rams were mustered, the sensors were removed and the rams separated from the mob. At 58 days after the mating period, not pregnant (dry), single and twin bearing ewes were identified by transabdominal ultrasound scanning, with a total of 64 ewes removed from the mob (63 dry and 1 dead).

At approximately 140 days from the start of the mating period, the remaining ewes (*n* = 352) were fitted with neck plates for easy identification and allocated randomly onto 22 small plots (1 ha), with approximately 16 ewes per plot for lambing. All plots were equipped with water troughs, 4 × 2 m straw bale shelter belts and feed troughs. Ewes were fed a lupin:oat (40:60) supplement once weekly and given *ad libitum* access to a calcium sulphate powder and molasses in addition to *ad libitum* oaten hay. Twice daily lambing rounds were conducted between mid-June and mid-July 2021 and lamb birth dates and birth type (single or multiple) were recorded.

Tissue samples (an ear punch) were collected from each ram prior to mating, each ewe prior to lambing and each lamb at marking. Tissue samples were processed to ascertain DNA parentage for the lambs and these results were used to adjust birth type (*n* = 40) and birth dates (*n* = 6) for some ewes recorded incorrectly.

### 2.2. Sensors

GT3X v4.4.0 sensors (ActiGraph, Pensacola, FL, USA) were used to measure proximity via Bluetooth. Interactions between the ewes and rams were recorded at 30 hz over an approximate 1–15 m range (range varies according to weather conditions and interference) every minute. A maximum of ten interactions per sensor were recorded each minute (e.g., 10 ewes within range of a ram equals 10 ewe/ram interactions; if there were more than ten ewes within about 15 m of the ram then the ten closest ewes to the ram were recorded for that minute).

Ewes were fitted with three beacons on a collar (total of three sensors per ewe), while rams were fitted with four receivers on two collars (total of four sensors; receivers use more battery requiring an additional sensor). Ewe sensors were pre-programmed to activate on three selected dates, to allow continuous recording between sensors to cater for battery life. Ram sensors were also programmed to consider battery life but were programmed to activate on four selected dates. The sensors were wrapped in silicon tape to prevent weather damage and cable tied to collars that were fitted to the sheep in a handling race. Unique electronic identification numbers on the collars were paired with the matching electronic identification number on the animal ear tag using an XRS2 TruTest stick reader (Datamars, Banyo, Queensland, Australia).

No (or partial) data from day 28 to day 35 were recorded for all ewes and rams due to battery failures leaving 27 days of interaction data for analysis. The sensors fitted to two rams experienced battery failure on the 16th and 23rd day of mating, respectively, causing partial data loss. Data loss due to sensor failures also occurred for seven ewes that were removed from the dataset. Ewes were also removed from the dataset if they didn’t have a lambing date recorded (*n* = 6) or if their lambing date was 150 days or more after day 27 of the mating period (*n* = 22), resulting in a total of 317 ewes and their 429 lambs used in the analysis.

### 2.3. Data Analysis

Lamb birth dates were predicted from the ewe’s date of peak interactions (predicted date of oestrous) expressed for each ewe by a “Maximum daily interactions ratio”; the day on which the ewe had the highest ratio of interactions with the rams. The ratio (for each day and ewe combination) is calculated as the number of interactions between the ewe and all rams divided by the average number of interactions for all ewes on that day. For example, on day 8 the average number of interactions of a ewe with the rams was 169, while one of the ewes interacting 995 times giving it a ratio of 5.9. This ewe didn’t record a higher ratio on any other day and therefore day 8 was its predicted date of oestrous.

A gestation length of 150 days was used for all ewes as per the methods of Paganoni et al., [24], where gestation length was defined as the number of days between mating and lamb birth. This gestation length was added to each date of peak interactions ratio to estimate lambing dates. The accuracy of predicting lamb birth dates was calculated at ±3 (total 7 days) and ±6 (total 13 days).

## 3. Results

### 3.1. Ewe and Ram Interactions during the Mating Period

On average, a ram interacted with a ewe 22 times a day over the first 27 days of the mating period, with the average number of interactions ranging from 13 times a day (excluding battery failures) to 33 times a day. The average number of interactions increased over time, peaking at day 25 (Figure 1). Three plateaus in interactions occurred on days 1–10, 14–20 and 22–27. Short term declines in interactions were observed on days 10–13 and 21. A reduction in the minimum number of interactions achieved per ram/day occurred on day 16 (5 interactions) and day 23 of mating (11 interactions), respectively, due to partial data capture for two rams (see methods). On any given day, the variation in the average number of interactions for individual rams was small (approximately five), representing no distinct differences in social behaviour between rams.

The maximum daily temperature recorded during the mating period peaked at 37.5 °C on day 9 and generally declined toward the end of mating, reaching just 18.6 °C on day 27. Rain events occurred on days 10 and 11 with 4.4 and 0.2 mm of rain, respectively (Figure 1).

### 3.2. Distribution of Lambing Date

For the 317 ewes used in the analysis, lambing commenced on the 5th of June (day 4) and ended on the 5th of July (day 26), where day 1 of lambing refers to 150 days after the rams were introduced. A total of 429 lambs were born including 206 singles and 223 multiples (from 111 ewes). The greatest number of lambs were born on day 16 (*n* = 27) (Figure 2, solid line).

### 3.3. Predicting Lambing Date

The predicted lambing date was within ±3 and ±6 days of the actual lambing date for 84% and 91% of the ewes, respectively (Figure 3). There was no significant difference in accuracy for single bearing verses multiple bearing ewes. For example, predictions were within ±6 days of the actual lambing date for 92% of single bearing ewes compared to 88% of multiple bearing ewes (*p* = 0.32).

The maximum number of daily interactions for a single ewe (with all 9 rams) ranged from 244 interactions/day to 1457 interactions/day. The percentage of lambing dates predicted correctly increased as the number of interactions between rams and ewes increased (Table 1). In all, 94% and 99% of ewes with a maximum daily interactions ratio above 3 (*n* = 104) had predicted lambing dates within ±3 and ±6 days respectively. While the accuracy decreased with lower ratios, the lambing dates for over half of the small group of ewes with a ratio below 1.5 (*n* = 23) were still estimated within ±3 days.

### 3.4. Gestation Length

The prediction accuracy was the highest for a gestation length of 149 or 150 days and either of these would be an acceptable choice (Table 2). Prediction accuracy decreases substantially (at ±1 and ±3 days) when the gestation length is reduced to 148 days or increased to 151 days.

### 3.5. DNA Results

The ram with maximum interactions on the estimated day of conception was only the successful sire for 16% of lambs born (Table 3). Two rams were more dominant, siring 61% of all lambs born (Table 3). One ram did not sire any progeny and another two sired only 3 lambs each (1%). Of the sets of twins born (*n* = 110), 37% (*n* = 41) had differing sires.

## 4. Discussion

Using proximity sensors to identify the time of oestrus from ewe–ram interactions during mating, and assuming a gestation period of 150 days allowed us to precisely predict the lambing date of Merino ewes under commercial production conditions. In all, 84% and 91% of actual lambing dates for all ewes were predicted within ±3 and ±6 days, respectively, for 24-month-old first parity ewes. The results therefore support the hypothesis that measuring ewe–ram interactions during the mating period can predict lambing dates precisely for ewes run under commercial production conditions. The capacity to predict lambing dates will enable farmers to manage cohorts of ewes with similar gestational ages to be managed more precisely during mid and late pregnancy and lambing to optimise ewe nutrition, lamb birthweights, access to shelter and mob sizes to increase the overall survival of lambs. Lambing cohorts of ewes over 7 to 13 days (±3 to ±6) could also increase the practicality of strategies to improve lamb survival, such as feeding starch supplements during the last seven days of pregnancy to boost colostrum production [26] and simplify the management and marketing of lambs that are more uniform for age and weight. Finally, as described by Paganoni et al. [24], more precise estimates of lambing date and hence lamb age could also improve the accuracy of estimated breeding values, especially those relating to early life growth potential.

Lamb birth dates were predicted from the day on which the ewe had the highest ratio of interactions, where the ratio was calculated as the number of interactions between the ewe and all rams divided by the average number of interactions for all ewes on that day. Initially a ewe’s maximum daily interactions ratio was calculated based on the interactions of the ewe with individual rams. However, this method was less accurate, and the DNA parentage data showed the ram corresponding with the maximum daily interactions ratio was the sire for only 16% of all progeny. This suggests that the ewes mated typically with multiple rams during oestrus, which is not entirely surprising given that Clarke et al. [27] recently reported than more than 50% of multiple born lambs from syndicate mated Merino ewes were from different sires. Our results were similar, with 37% of multiple born lambs having differing sires. We therefore concluded that, whilst proximity sensors could accurately predict the time of lambing, they could not be used to indicate the sire of the progeny.

The present study, together with Paganoni et al. [24], indicate that using proximity sensors to identify the time of conception to predict the time of lambing is likely to be less variable and more accurate than other methods, such as foetal aging using transabdominal ultrasound scanning. Bunter et al. [15] used a highly competent ultrasound technician and achieved an accuracy of 57% within 3.5 days and 92% within 7 days of actual lambing date. Likewise, Robertson et al. [19] reported that between 86 and 94% of ewes lambed within ±5-days of the predicted lambing date when the foetuses were 10.5 to 12 weeks old at scanning; however, when the foetuses were 8.5 to 10 weeks old, the accuracy dropped to 23 to 44%. Neither of these studies reported the accuracy of the predicting time of lambing for single and twin bearing ewes, but it is well recognized that it is more difficult to assess the fetal age of multiples than singles using ultrasound scanning due to reduced visibility and variation in size between fetuses [14,20,21]. In contrast, the accuracy of predicting time of lambing using proximity sensors was not influenced by litter size. Given that multiple born lambs are more vulnerable to the effects of ewe nutrition, birthweight, mob size and shelter [1,2,9,10,28], using proximity sensors could benefit the management of multiple bearing ewes both pre- and post-lambing.

Comparatively, the predicted lambing dates in this experiment were less precise than the 100% reported by Paganoni et al. [24]. Part of these differences can be explained by Paganoni et al. [24] excluding about 10% of ewes with the lowest interactions ratios, whereas in the current experiment we included these ewes despite their lower accuracy. Three other factors are likely to contribute to the small difference in accuracy between studies. First, this study investigated the interactions of younger, inexperienced (first parity/maiden) ewes with rams, and younger or less experienced ewes are known to have a shorter oestrous cycle and/or reduced ‘sexual interest’ in comparison to older ewes [29,30,31]. This would contribute to a reduced expression of oestrous behaviours and a subsequent reduction in the number of interactions compared to the adult ewes used by Paganoni et al. [24]. Second, this study was conducted across a larger paddocks (50 vs. 1 hectare) with greater samples sizes (327 vs. 55). Considering the gregarious nature of sheep, it is acceptable to expect a reduction in the total number of interactions between ewes and rams with a reduced stocking rate [32,33]. Third, an increase in the number of rams mated with ewes contributes to an increase in competition during oestrous [34], therefore leading to less interactions per ewe, per ram. Thus, the comparison between studies would suggest that the accuracy of predicted lambing dates using proximity sensors will be greater for older multiparous ewes and using fewer rams per ewe within smaller mating paddocks.

Extreme weather events are known to cause changes in the grazing behaviours of ewes [35], and consequently may affect interactions between ewes and rams, potentially reducing the accuracy of lambing date predictions using proximity sensors. A small rainfall event appeared to reduce ewe–ram interactions on day 10 of mating. This was coupled with a decrease in temperature, although the average number of ewe–ram interactions increased over the mating period which coincided with decreasing daily maximum temperatures. In this study, the effects of temperature were confounded with the effects of paddock and social acclimatisation between animals. Further work is needed to quantify the effects of varying environmental conditions during the mating period on the effectiveness of proximity sensors for predicting time of lambing.

## 5. Conclusions

This study confirms that sensors to determine the proximity between ewes and rams during mating can be used to accurately identify the time of oestrus in ewes and therefore predict lambing dates under commercial production conditions. This method offers considerable advantages in comparison to current methods to predict lambing dates, such as foetal ageing, in that it is not reliant on operator skill and is less sensitive to litter size. The capacity to predict lambing dates will enable farmers to manage cohorts of ewes with similar gestational ages to be managed more precisely during late pregnancy and lambing to improve overall survival of lambs, as well as opportunities to improve lamb performance and the accuracy of breeding values. Further improvements in prediction accuracy may be achieved when mating older, multiparous ewes in smaller mating paddocks.

## Figures and Tables

**Figure 1 animals-12-01707-f001:**
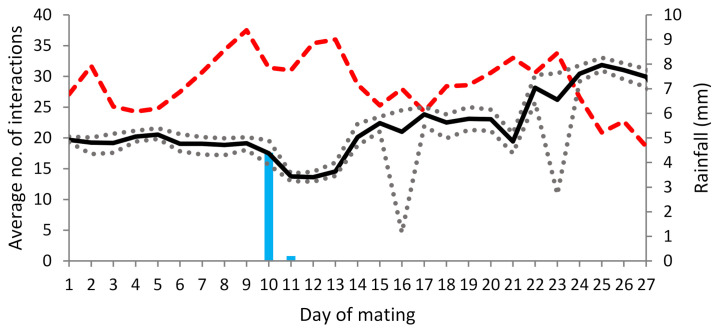
The average number of interactions for all rams across all ewes each day (**-**), the maximum and minimum number of interactions for all rams across all ewes each day (…), the maximum temperature °C (- - -) and rainfall (

) recorded over 27 days grazing dry stubble.

**Figure 2 animals-12-01707-f002:**
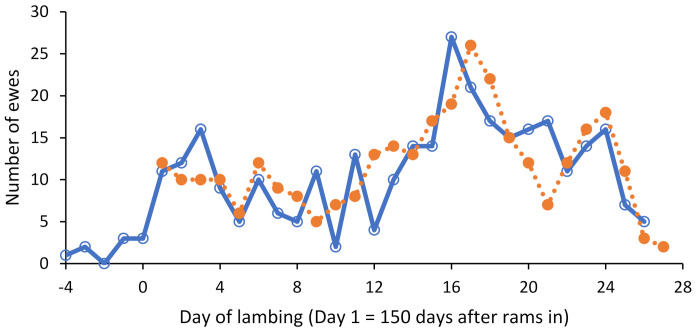
Daily frequency of ewe actual day of lambing (solid line) compared to predicted day of lambing (dotted line) based on the maximum daily interactions ratio (*n* = 317).

**Figure 3 animals-12-01707-f003:**
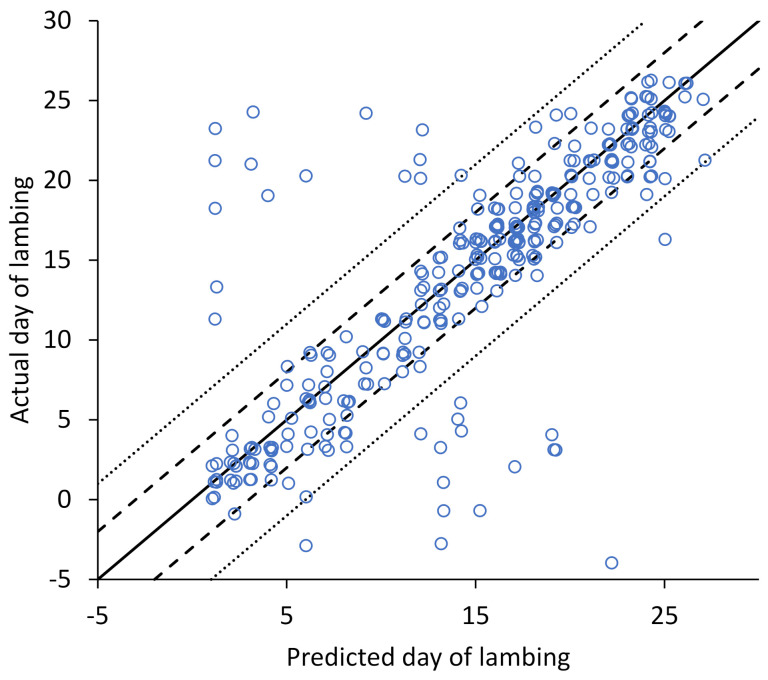
The predicted day of lambing based on 150 days after the maximum daily interactions ratio compared to the actual day of lambing. A small random jitter of less than ±0.3 was added so ewes with the same combination are slightly offset. Day 0 of lambing is 150 days after rams went in with the ewes. The solid line represents when the predicted day of lambing is the actual day of lambing. The dashed and dotted lines are ±3 and ±6 days respectively.

**Table 1 animals-12-01707-t001:** The percentage of predicted lambing date within 3 and 6 days of the actual lambing date for different ranges of maximum daily interactions ratio.

Maximum Daily	Number	Accuracy	
Interactions Ratio	of Ewes	+/− 3 Days	+/− 6 Days
>3	104	94%	99%
2.5–3	62	90%	97%
2–2.5	69	83%	88%
1.5–2	59	73%	85%
<1.5	23	52%	57%
All	317	84%	91%

**Table 2 animals-12-01707-t002:** The effect of using different gestation lengths on the proportion of ewes lambing within ±1, ±3 and ±6 days of the date predicted.

Gestation	Accuracy		
Length (Days)	+/− 1 Days	+/− 3 Days	+/− 6 Days
147	25.2%	66.9%	90.2%
148	41.6%	78.2%	90.2%
149	54.9%	84.9%	90.2%
150	52.1%	83.9%	90.5%
151	40.1%	77.6%	89.9%
152	21.8%	63.4%	89.3%

**Table 3 animals-12-01707-t003:** The number and percentage of lambs sired by the ram that had the maximum number of interactions with the dam on her estimated day of conception.

Sire	No. of Progeny	% of All Progeny	No. of Progeny from Ram with Maximum Interactions	% of All Progeny from Ram with Maximum Interactions
20160024	49	11%	7	2%
20160147	45	11%	7	2%
20160613	0	0%	0	0%
20160777	135	32%	13	3%
20160810	126	29%	19	4%
20161129	3	1%	0	0%
20164120	3	1%	2	0%
20164165	47	11%	12	3%
20164209	20	5%	10	2%
Total	*428	100%	70	16%

* One lamb did not return a sire result from DNA.

## Data Availability

The datasets generated and/or analysed during the current study are not publicly available but are available from the corresponding author on reasonable request pending permission from project funders.

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
