# Peer review of "Interactions between Ewes and Rams during Mating Can Be Used to Predict Lambing Dates Accurately, but Not Sire"

_animals, 2022, doi:10.3390/ani12131707_

Round 1
Reviewer 1 Report
The present study aimed at using proximity sensors to predict lambing day in young ewes, with the final goal of reducing lambs' mortality. the idea is interesting and feasible, although economic prospect regarding the employment of these devices is not discussed. The paper is well-written, and clear and the study design is appropriate and entirely described. Discussion is coherent with results. I have only a few minor points to discuss:
- INTRODUCTION, line 64: to ESTIMATE fetal age?
- INTRODUCTION, line83: please cite paper (24) otherwise saying this work may be confusing with the present work
- paragraph 2.1.: how hold were the rams? please add data. Did you perform a breeding soundness evaluation of these rams?
- paragraph 2.2, line 117: WERE recorded? (otherwise, the sentence is incomplete)
- figure 1: please add caption within the figure to indicate you are using °C for temperature
- discussion, line241: you are mentioning split paternity: why you didn't check for split paternity in this work? since DNA data are available please discuss if split paternity is present and in which %
Author Response
Referee 1:
Point 1: INTRODUCTION, line 64: to ESTIMATE fetal age?
Response 1: Amended L64
Point 2: INTRODUCTION, line83: please cite paper (24) otherwise saying this work may be confusing with the present work
Response 2: Amended L83
Point 3: paragraph 2.1.: how hold were the rams? please add data. Did you perform a breeding soundness evaluation of these rams?
Response 3: Information added L94
Point 4: paragraph 2.2, line 117: WERE recorded? (otherwise, the sentence is incomplete)
Response 4: Amended L119
Point 5: figure 1: please add caption within the figure to indicate you are using °C for temperature
Response 5: Amended L174
Point 6: discussion, line241: you are mentioning split paternity: why you didn't check for split paternity in this work? since DNA data are available please discuss if split paternity is present and in which %
Response 6: Split paternity added to results L216 and discussion L247-248
Reviewer 2 Report
The aim of this paper was to evaluate the use of proximity sensors during mating to predict lambing dates. The hypothesis is that by measuring the interactions between ewes and rams during mating, as an indicator of conception, the lambing dates of ewes can be accurately predicted. Three hundred seventeen inexperienced Merino ewes and 9 experienced rams under commercial production conditions were used. The mating period considered was only 27 days because sensors’ battery failed from day 27 to day 35 of mating. Ninety one percent of lambing dates were predicted within ±6 days of the actual birth date of lambs and 84% of lambing dates were predicted within ±3 days. DNA parentage data showed the ram corresponding with the maximum daily interaction ratio was the sire for only 16% of all progeny, suggesting they could not be used to indicate the sire of the progeny. In conclusion, the use of proximity sensors during mating was an effective means of predicting lambing dates and there was no significant difference in accuracy for single bearing verses multiple bearing ewes. This could help improve ewe management during pregnancy and lambing and the subsequent survival and performance of their lambs.
The topic of the paper is important because ewes often lamb over extended periods so the level of nutrition during pregnancy and lambing maybe sub-optimal for ewes that conceived at the beginning or end of the mating period. The early prediction of lambing dates for ewes could improve their management at critical time points during pregnancy (during the end of pregnancy) and throughout lambing. Predicting lambing dates would allow cohorts of ewes with similar gestational ages to be managed more precisely in order to achieve targets for ewe nutrition, feed on offer, mob sizes and access to shelter to improve lamb survival. Furthermore, other important topics are discussed in the paper, among them: lamb mortality, mating period, ultrasound scanning, gestation length and behaviour of sheep and rams during mating.
The manuscript is clear, written in an appropriate way, relevant for the field and presented in a well-structured manner. The English language is appropriate and understandable. The manuscript’s results are reproducible based on the details given in the methods section. The analyses and results were presented appropriately, the figures and tables are appropriate and the data is easy to interpret and understand. The results were interpreted appropriately, they are significant and all conclusions were justified and supported by the results. The study is scientifically and technically adequate and the experimental design is appropriate to test the hypothesis. The analyses were performed with high technical standards and the data is robust enough to draw conclusions. The cited references are relevant and they do not include an excessive number of self-citations. The ethics statements and data availability statements are adequate. The aim of the study is original and well-defined and the results make an original contribution to the current knowledge. The work fits the journal scope and the conclusions are interesting for the readership of the journal and the paper will attract a wide readership.
Overall recommendation: Accept after Minor Revisions.
Specific comments referring to line numbers, tables or figures:
Lines 23, 56 & 57: …for ewes that conceived on their second or third oestrus cycle.
I wouldn't be that specific; I would say at the beginning or end of the service because this depends on the date of service, the forage supply, the weather conditions of the year, etc.
Lines 92 to 112: Please add the following information if it is possible: Were the rams clinically normal? Have they had a clinical examination of testis and epididymis? Were the rams seronegative to Brucella ovis? Didn’t you have abortions between the ultrasound scanning and the lambing period?
Line 95: -33°69’S and 117°57’E: Consider revision.
Lines 96 to 98: All ewes and rams were fitted with proximity sensors and grazed together over the mating period, before being mustered, sensors removed and rams separated from the mob. Please rephrase this sentence for better understanding.
Line 109: Tissue samples were collected from each ram prior to mating... Please add the type of tissue sampled.
Lines 111 & 112: … for a small number of ewes. Please rephrase for better understanding. How many ewes and why?
Line 114: ActiGraph GT3X sensors (v4.4.0 Pensacola Florida USA). Replace by: GT3X v4.4.0 sensors (ActiGraph, Pensacola, Florida, USA)
Lines 140 to 144: Lamb birth dates were predicted from the date of peak interactions (predicted date of oestrous) expressed for each ewe by a “Maximum daily interactions ratio”; the day on which the ewe had the highest ratio of interactions, where the ratio (for each day and ewe combination) is calculated as the number of interactions between the ewe and all rams divided by the average number of interactions for all ewes on that day. Please rephrase this paragraph for better understanding. Can you explain it also with an example?
Figure 1: There is not a reference of Temperature in the ordinate (Y) axis. It is confusing with the values of number of interactions.
Lines 168 & 169: …the maximum temperature (- - - (in red color if it is possible for better understanding))
Line 170: 3.2 . Distribution of lambing date. There is a space between the number 2 and the point.
Lines 179 & 180: The predicted lambing date was within ±3 and ±6 days of the actual lambing date for 84% and 91% of the ewes, respectively (Figure 3).
Lines 35, 147, 177, 185, 194, 199, 236, 238 Table 1: interaction ratio: be consistent and replace by interactions ratio (in plural), as explained in M&M section (line 141).
Lines 190 to 197: I would join the sentences in a single paragraph.
Lines 209 to 211: Two rams were more dominant, siring 61% of all lambs born (Table 3). One ram did not sire any progeny and another two sired only 3 lambs each (1%). These results, despite being a complementary information, are also very interesting because the reproductive capacity of each ram was identified in a direct and objective manner. You should discuss them in the corresponding section (Discussion). Is this technique (DNA parentage) not commonly used in commercial production conditions because of its cost? Why are there 2 rams that sire so few lambs and one that did not sire any progeny? Are them infertile? Subordinates? With low libido?
Table 3: Because of the difference of progeny sired between rams it would be interesting to add more characteristics of each ram, like testicular size, testicular tone, semen quality or reproductive capacity if it is possible. This information could be added in this table, in another table in M&M section or as supplementary material.
Line 226: …ewes over 7 to 13 days… 7 to 13 days prepartum? If they are not prepartum days I do not understand the sentence.
Line 316: Acknowledgments: Acknowledgements: Delete repeated word.
Author Response
Referee 2:
Point 1: Lines 23, 56 & 57: …for ewes that conceived on their second or third oestrus cycle.
I wouldn't be that specific; I would say at the beginning or end of the service because this depends on the date of service, the forage supply, the weather conditions of the year, etc.
Response 1: Amended L23-24 and L56-57
Point 2: Lines 92 to 112: Please add the following information if it is possible: Were the rams clinically normal? Have they had a clinical examination of testis and epididymis? Were the rams seronegative to Brucella ovis? Didn’t you have abortions between the ultrasound scanning and the lambing period?
Response 2: Information added L94
Point 3: Line 95: -33°69’S and 117°57’E: Consider revision.
Response 3: Corrected L95
Point 4: Lines 96 to 98: All ewes and rams were fitted with proximity sensors and grazed together over the mating period, before being mustered, sensors removed and rams separated from the mob. Please rephrase this sentence for better understanding.
Response 4: Amended L97-98
Point 5: Line 109: Tissue samples were collected from each ram prior to mating... Please add the type of tissue sampled.
Response 5: Amended L110
Point 6: Lines 111 & 112: … for a small number of ewes. Please rephrase for better understanding. How many ewes and why?
Response 6: Amended L112-L113
Point 7: Line 114: ActiGraph GT3X sensors (v4.4.0 Pensacola Florida USA). Replace by: GT3X v4.4.0 sensors (ActiGraph, Pensacola, Florida, USA)
Response 7: Amended L115
Point 8: Lines 140 to 144: Lamb birth dates were predicted from the date of peak interactions (predicted date of oestrous) expressed for each ewe by a “Maximum daily interactions ratio”; the day on which the ewe had the highest ratio of interactions, where the ratio (for each day and ewe combination) is calculated as the number of interactions between the ewe and all rams divided by the average number of interactions for all ewes on that day. Please rephrase this paragraph for better understanding. Can you explain it also with an example?
Response 8: Amended L141-149
Point 9: Figure 1: There is not a reference of Temperature in the ordinate (Y) axis. It is confusing with the values of number of interactions.
Response 9: Amended L74
Point 10: Lines 168 & 169: …the maximum temperature (- - - (in red color if it is possible for better understanding))
Response 10: Defined in the figure key L173-174
Point 11: Line 170: 3.2 . Distribution of lambing date. There is a space between the number 2 and the point.
Response 11: Amended L175
Point 12: Lines 179 & 180: The predicted lambing date was within ±3 and ±6 days of the actual lambing date for 84% and 91% of the ewes, respectively (Figure 3).
Response 12: Amended L185
Point 13: Lines 35, 147, 177, 185, 194, 199, 236, 238 Table 1: interaction ratio: be consistent and replace by interactions ratio (in plural), as explained in M&M section (line 141).
Response 13: Corrected throughout paper to “interactions ratio”
Point 14: Lines 190 to 197: I would join the sentences in a single paragraph.
Response 14: Amended L195-202.
Point 15: Lines 209 to 211: Two rams were more dominant, siring 61% of all lambs born (Table 3). One ram did not sire any progeny and another two sired only 3 lambs each (1%). These results, despite being a complementary information, are also very interesting because the reproductive capacity of each ram was identified in a direct and objective manner. You should discuss them in the corresponding section (Discussion). Is this technique (DNA parentage) not commonly used in commercial production conditions because of its cost? Why are there 2 rams that sire so few lambs and one that did not sire any progeny? Are them infertile? Subordinates? With low libido?
Response 15: We agree that while the above points are possible and interesting they are not core (tertiary) to the hypothesis being tested and would distract from the key results, particularly given limited measures of ram fertility were measured in our design
Point 16: Table 3: Because of the difference of progeny sired between rams it would be interesting to add more characteristics of each ram, like testicular size, testicular tone, semen quality or reproductive capacity if it is possible. This information could be added in this table, in another table in M&M section or as supplementary material.
Response 16: We agree, however these measurements were not made/part of the experimental design so commenting on potential reasons for differing ram performance is difficult/limited.
Point 17: Line 226: …ewes over 7 to 13 days… 7 to 13 days prepartum? If they are not prepartum days I do not understand the sentence.
Response 17: Extra information added L231
Point 18: 316: Acknowledgments: Acknowledgements: Delete repeated word.
Response 18: Amended L332
